# Instance-Dependent Partial Label Learning

**Ning Xu, Congyu Qiao, Xin Geng,**\* and **Min-Ling Zhang**
School of Computer Science and Engineering, Southeast University, Nanjing 210096, China
MOE Key Laboratory of Computer Network and Information Integration, Ministry of Education, China
`{xning, qiaocy, xgeng, zhangml}@seu.edu.cn`

## Abstract

Partial label learning (PLL) is a typical weakly supervised learning problem, where each training example is associated with a set of *candidate* labels among which only one is true. Most existing PLL approaches assume that the incorrect labels in each training example are randomly picked as the candidate labels. However, this assumption is not realistic since the candidate labels are always instance-dependent. In this paper, we consider instance-dependent PLL and assume that each example is associated with a latent *label distribution* constituted by the real number of each label, representing the degree to each label describing the feature. The incorrect label with a high degree is more likely to be annotated as the candidate label. Therefore, the latent label distribution is the essential labeling information in partially labeled examples and worth being leveraged for predictive model training. Motivated by this consideration, we propose a novel PLL method that recovers the label distribution as a label enhancement (LE) process and trains the predictive model iteratively in every epoch. Specifically, we assume the true posterior density of the latent label distribution takes on the variational approximate Dirichlet density parameterized by an inference model. Then the evidence lower bound is deduced for optimizing the inference model and the label distributions generated from the variational posterior are utilized for training the predictive model. Experiments on benchmark and real-world datasets validate the effectiveness of the proposed method. Source code is available at `https://github.com/palm-ml/valen`.

## 1 Introduction

Partial label learning (PLL) deals with the problem where each training example is associated with a set of candidate labels, among which only one label is valid [7, 5, 37]. Due to the difficulty in collecting exactly labeled data in many real-world scenarios, PLL leverages inexact supervision instead of exact labels. The need to learn from the inexact supervision leads to a wide range of applications for PLL techniques, such as web mining [25], multimedia content analysis [38, 4], ecoinformatics [24, 31], etc.

To accomplish the task of learning from partial label data, many approaches have been proposed. Identification-based PLL approaches [17, 28, 24, 5, 37] regard the ground-truth label as a latent variable and try to identify it. Average-based approaches [15, 7, 40] treat all the candidate labels equally and average the modeling outputs as the prediction. For confidence-based approaches [8, 34, 42], the confidence of each label is estimated instead of identifying the ground-truth label. These approaches always adopt the randomly picked candidate labels to corrupt benchmark data into partially labeled version despite having no explicit generation process of candidate label sets. To depict the instance-independent generation process of candidate label sets, Feng [9] proposes a statistical model and deduces a risk-consistent method and a classifier-consistent method. Under the

---

\*Corresponding author

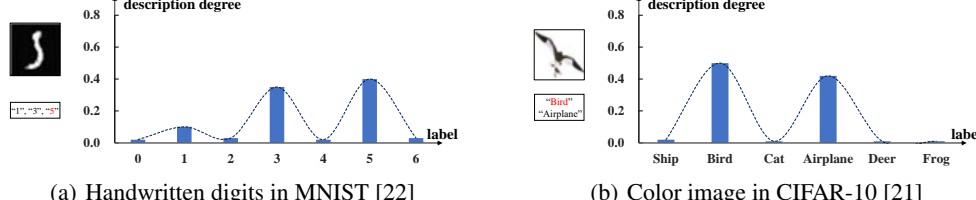

(a) Handwritten digits in MNIST [22]  (b) Color image in CIFAR-10 [21]

Figure 1: The examples about the latent label distributions for partial label learning. The candidate labels are in the box and the red one is valid.

same generation process, another classifier-consistent risk estimator is proposed for deep model and stochastic optimizers [26].

The previous methods assume that the candidate labels are randomly sampled with the uniform generating procedure [26, 9], which is commonly adopted to corrupt benchmark datasets into partially labeled versions in their experiments. However, the candidate labels are always instance-dependent (feature-dependent) in practice as the incorrect labels related to the feature are more likely to be picked as candidate label set for each instance. These methods usually do not perform as well as expected due to the unrealistic assumption on the generating procedure of candidate label sets.

In this paper, we consider instance-dependent PLL and assume that each instance in PLL is associated with a latent *label distribution* [33, 35, 11] constituted by the real number of each label, representing the degree to each label describing the feature. Then, the incorrect label with a high degree in the latent label distribution is more likely to be annotated as the candidate label. For example, the candidate label set of the handwritten digits in Figure 1(a) contains "1", "3" and "5", where "1" and "3" are not ground-truth but selected as candidate labels due to their high degrees in the latent label distribution of the instance. The object in Figure 1(b) is annotated with "bird" and "airplane" as the degrees of these two labels are much higher than others in the label distribution. The intrinsical ambiguity increases the difficulty of annotating, which leads to the result that annotators pick the candidate labels with high degrees in the latent label distribution of each instance instead of annotating the ground-truth label directly in PLL. Therefore, the latent label distribution is the essential labeling information in partially labeled examples and worth being leveraged for predictive model training.

Motivated by the above consideration, we deal with the PLL problem from two aspects. First, we enhance the labeling information by recovering the latent label distribution for each training example as a label enhancement process [33, 35]. Second, we run label enhancement and train the predictive model with recovered label distributions iteratively. The proposed method named VALEN, i.e., *VAriational Label ENhancement for instance-dependent partial label learning*, uses the candidate labels to initialize the predictive model in the warm-up training stage, then recovers the latent label distributions via inferring the variational posterior density parameterized by an inference model with the deduced evidence lower bound, and trains the predictive model with a risk estimator by leveraging the candidate labels as well as the label distributions. Our contributions can be summarized as follows:

- We for the first time consider the instance-dependent PLL and assume that each partially labeled example is associated with a latent label distribution, which is the essential labeling information and worth being recovered for predictive model training.
- We infer the posterior density of the latent label distribution via taking on the approximate Dirichlet density parameterized by an inference model and deduce the evidence lower bound for optimization, in which the topological information and the features extracted from the predictive model are leveraged.
- We train predictive model with a proposed empirical risk estimator by leveraging the candidate labels as well as the label distributions. We iteratively recover the latent label distributions and train the predictive model in every epoch. After the network has been fully trained, the predictive model can perform predictions for future test examples alone.

Experiments on the corrupted benchmark datasets and real-world PLL datasets validate the effectiveness of the proposed method.

## 2 Proposed Method

First of all, we briefly introduce some necessary notations. Let $\mathcal{X} = \mathbb{R}^q$ be the $q$-dimensional instance space and $\mathcal{Y} = \{y_1, y_2, ..., y_c\}$ be the label space with $c$ class labels. Given the PLL training set $\mathcal{D} = \{(\boldsymbol{x}_i, S_i)|1 \leq i \leq n\}$ where $\boldsymbol{x}_i$ denotes the $q$-dimensional instance and $S_i \subseteq \mathcal{Y}$ denotes the candidate label set associated with $\boldsymbol{x}_i$. Note that $S_i$ contains the correct label of $\boldsymbol{x}_i$ and the task of PLL is to induce a multi-class classifier $f : \mathcal{X} \mapsto \mathcal{Y}$ from $\mathcal{D}$. For each PLL training example $(\boldsymbol{x}_i, S_i)$, we use the logical label vector $\boldsymbol{l}_i = [l_i^{y_1}, l_i^{y_2}, \ldots, l_i^{y_c}]^\top \in \{0, 1\}^c$ to represent whether $y_j$ is the candidate label, i.e., $l_i^{y_j} = 1$ if $y_j \in S_i$, otherwise $l_i^{y_j} = 0$. The label distribution of $\boldsymbol{x}_i$ is denoted by $\boldsymbol{d}_i = [d_i^{y_1}, d_i^{y_2}, \ldots, d_i^{y_c}]^\top \in [0, 1]^c$ where $\sum_{j=1}^c d_i^{y_j} = 1$. Then $\mathbf{L} = [\boldsymbol{l}_1, \boldsymbol{l}_2, \ldots, \boldsymbol{l}_n]$ and $\mathbf{D} = [\boldsymbol{d}_1, \boldsymbol{d}_2, \ldots, \boldsymbol{d}_n]$ represent the logical label matrix and label distribution matrix, respectively.

### 2.1 Overview

To deal with PLL problem, we iteratively recover the latent label distribution for each example $\boldsymbol{x}$ and train the predictive model by leveraging the recovered label distribution. We start with a warm-up period, in which we train the predictive model with the PLL minimal loss [26]. This allows us to attain a reasonable predictive model before it starts fitting incorrect labels. After the warm-up period, the features extracted from the predictive model can help for recovering the latent label distribution. Benefited from the essential labeling information in the recovered label distribution, the performance of the predictive model could be further improved.

VALEN implements label enhancement and classifier training iteratively in every epoch. In label enhancement, we assume the true posterior density of the latent label distribution takes on the variational approximate Dirichlet density parameterized by an inference model. Then the evidence lower bound is deduced for optimizing the inference model and the label distributions can be generated from the variational posterior. In classifier training, the predictive model is trained by leveraging the recovered label distributions and candidate labels with an empirical risk estimator. After the models has been fully trained, the predictive model can perform prediction for future test instances alone.

### 2.2 Warm-up Training

The predictive model $\boldsymbol{\theta}$ is trained on partially labeled examples by minimizing the following PLL minimal loss function [26]:

$$\mathcal{L}_{min} = \sum_{i=1}^n \min_{y_j \in S_i} \ell(f(\boldsymbol{x}_i), \boldsymbol{e}^{y_j}), \tag{1}$$

where $\ell$ is cross-entropy loss and $\boldsymbol{e}^{\mathcal{Y}} = \{\boldsymbol{e}^{y_j} : y_j \in \mathcal{Y}\}$ denotes the standard canonical vector in $\mathcal{R}^c$, i.e., the $j$-element in $\boldsymbol{e}^{y_j}$ equals 1 and others equal 0. Similar to [26], the min operator in Eq. (1) is replaced by using the current predictions for slightly weighting on the possible labels in warm-up training. Then we could extract the feature $\boldsymbol{\phi}$ of each $\boldsymbol{x}$ via using the predictive model.

### 2.3 Label Enhancement

We assume that the prior density $p(\boldsymbol{d})$ is a Dirichlet with $\hat{\boldsymbol{\alpha}}$, i.e., $p(\boldsymbol{d}) = Dir(\boldsymbol{d} \mid \hat{\boldsymbol{\alpha}})$ where $\hat{\boldsymbol{\alpha}} = [\varepsilon, \varepsilon, \ldots, \varepsilon]^\top$ is a $c$-dimensional vector with a minor value $\varepsilon$. Then we let the prior density $p(\mathbf{D})$ be the product of each Dirichlet

$$p(\mathbf{D}) = \prod_{i=1}^n Dir(\boldsymbol{d}_i|\hat{\boldsymbol{\alpha}}). \tag{2}$$

We consider the topological information of the feature space, which is represented by the affinity graph $G = (V, E, \mathbf{A})$. Here, the feature vector $\boldsymbol{\phi}_i$ of each example could be extracted from the predictive model $\boldsymbol{\theta}$ in current epoch, $V = \{\boldsymbol{\phi}_i \mid 1 \leq i \leq n\}$ corresponds to the vertex set consisting of feature vectors, $E = \{(\boldsymbol{\phi}_i, \boldsymbol{\phi}_j) \mid 1 \leq i \neq j \leq n\}$ corresponds to the edge set, and a sparse adjacency matrix $\mathbf{A} = [a_{ij}]_{n \times n}$ can be obtained by

$$a_{ij} = \begin{cases} 1 & \text{if } \boldsymbol{\phi}_i \in \mathcal{N}(\boldsymbol{\phi}_j) \\ 0 & \text{otherwise} \end{cases}, \tag{3}$$

where $\mathcal{N}(\phi_j)$ is the set for $k$-nearest neighbors of $\phi_j$ and the diagonal elements of $\mathbf{A}$ are set to 1.

Let features matrix $\mathbf{\Phi} = [\phi_1, \phi_2, \ldots, \phi_n]$, adjacency matrix $\mathbf{A}$ and logical labels $\mathbf{L}$ be observed matrix, VALEN aims to infer the posterior density $p(\mathbf{D}|\mathbf{L}, \mathbf{\Phi}, \mathbf{A})$. As the computation of the exact posterior density $p(\mathbf{D}|\mathbf{L}, \mathbf{\Phi}, \mathbf{A})$ is intractable, a fixed-form density $q(\mathbf{D}|\mathbf{L}, \mathbf{\Phi}, \mathbf{A})$ is employed to approximate the true posterior. We let the approximate posterior be the product of each Dirichlet parameterized by a vector $\boldsymbol{\alpha}_i = [\alpha_i^1, \alpha_i^2, \ldots, \alpha_i^c]^\top$:

$$q_{\boldsymbol{w}}(\mathbf{D} \mid \mathbf{L}, \mathbf{\Phi}, \mathbf{A}) = \prod_{i=1}^{n} Dir(\boldsymbol{d}_i | \boldsymbol{\alpha}_i). \tag{4}$$

Here, the parameters $\mathbf{\Delta} = [\boldsymbol{\alpha}_1, \boldsymbol{\alpha}_2, \ldots, \boldsymbol{\alpha}_n]$ are outputs of the inference model parameterized by $\boldsymbol{w}$, which is defined as a two-layer GCN [20] by $\mathrm{GCN}(\mathbf{L}, \mathbf{\Phi}, \mathbf{A}) = \tilde{\mathbf{A}} \, \mathrm{ReLU}\left(\tilde{\mathbf{A}} \mathbf{Z} \mathbf{W}_0\right) \mathbf{W}_1$, with $\mathbf{Z} = [\mathbf{\Phi}; \mathbf{L}]$ and weight $\mathbf{W}_0, \mathbf{W}_1$. Here $\tilde{\mathbf{A}} = \hat{\mathbf{A}}^{-\frac{1}{2}} \mathbf{A} \hat{\mathbf{A}}^{-\frac{1}{2}}$ is the symmetrically normalized weight matrix where $\hat{\mathbf{A}}$ is the degree matrix of $\mathbf{A}$.

By following the Variational Bayes techniques, a lower bound on the marginal likelihood of the model is derived which ensures that $q_{\boldsymbol{w}}(\mathbf{D}|\mathbf{L}, \mathbf{\Phi}, \mathbf{A})$ is as close as possible to $p(\mathbf{D}|\mathbf{L}, \mathbf{\Phi}, \mathbf{A})$. For logical label matrix $\mathbf{L}$, feature matrix $\mathbf{\Phi}$, and the corresponding $\mathbf{A}$, the log marginal probability can be decomposed as follows [1]:

$$\log p(\mathbf{L}, \mathbf{\Phi}, \mathbf{A}) = \mathcal{L}_{ELBO} + \mathrm{KL}[q_{\boldsymbol{w}}(\mathbf{D}|\mathbf{L}, \mathbf{\Phi}, \mathbf{A})||p(\mathbf{D}|\mathbf{L}, \mathbf{\Phi}, \mathbf{A})]. \tag{5}$$

where

$$\mathcal{L}_{ELBO} = \mathbb{E}_{q_{\boldsymbol{w}}(\mathbf{D}|\mathbf{L}, \mathbf{\Phi}, \mathbf{A})}[\log p(\mathbf{L}, \mathbf{\Phi}, \mathbf{A}|\mathbf{D})] - \mathrm{KL}[q_{\boldsymbol{w}}(\mathbf{D}|\mathbf{L}, \mathbf{\Phi}, \mathbf{A})||p(\mathbf{D})]. \tag{6}$$

Due to the non-negative property of KL divergence, the first term $\mathcal{L}_{ELBO}$ constitutes a lower bound of $\log p(\mathbf{L}, \mathbf{\Phi}, \mathbf{A})$, which is often called as evidence lower bound (ELBO), i.e., $\log p(\mathbf{L}, \mathbf{\Phi}, \mathbf{A}) \geq \mathcal{L}_{ELBO}$.

According to Eq. (2) and Eq. (4), the KL divergence in Eq. (6) can be analytically calculated as follows:

$$\begin{aligned} \mathrm{KL}\left(q_{\boldsymbol{w}}(\mathbf{D}|\mathbf{L}, \mathbf{\Phi}, \mathbf{A})||p(\mathbf{D})\right) = \sum_{i=1}^{n} &\left( \log \Gamma\left(\sum_{j=1}^{c} \alpha_i^j\right) - \sum_{j=1}^{c} \log \Gamma\left(\alpha_i^j\right) \right. \\ &\left. - \log \Gamma\left(c \cdot \varepsilon\right) + c \log \Gamma\left(\varepsilon\right) + \sum_{j=1}^{c} \left(\alpha_i^j - \varepsilon\right)\left(\psi\left(\alpha_i^j\right) - \psi\left(\sum_{j=1}^{c} \alpha_i^j\right)\right) \right). \end{aligned} \tag{7}$$

where $\Gamma(\cdot)$ and $\psi(\cdot)$ are Gamma function and Digamma function, respectively.

As the first part of Eq. (6) is intractable, we employ the implicit reparameterization trick [10] to approximate it by Monte Carlo (MC) estimation. Inspired by [20], we simply drop the dependence on $\mathbf{\Phi}$:

$$\begin{aligned} p(\mathbf{L} \mid \mathbf{A}, \mathbf{D}) &= \prod_{i=1}^{n} p(\boldsymbol{l}_i \mid \mathbf{A}, \mathbf{D}), \\ p(\mathbf{A} \mid \mathbf{D}) &= \prod_{i=1}^{n} \prod_{j=1}^{n} p(a_{ij} \mid \boldsymbol{d}_i, \boldsymbol{d}_j), \text{ with } p(a_{ij} = 1 \mid \boldsymbol{d}_i, \boldsymbol{d}_j) = s\left(\boldsymbol{d}_i^\top \boldsymbol{d}_j\right). \end{aligned} \tag{8}$$

Here, $s(\cdot)$ is the logistic sigmoid function. We further assume that $p(\boldsymbol{l}_i|\mathbf{A}, \mathbf{D})$ is a multivariate Bernoulli with probabilities $\boldsymbol{\tau}_i$. In order to simplify the observation model, $\mathbf{T}^{(m)} = [\boldsymbol{\tau}_1^{(m)}, \boldsymbol{\tau}_2^{(m)}, \ldots, \boldsymbol{\tau}_n^{(m)}]$ is computed from $m$-th sampling $\mathbf{D}^{(m)}$ with a three-layer MLP parameterized by $\boldsymbol{\eta}$. Then the first part of Eq. (6) can be tractable:

$$\begin{aligned} \mathbb{E}_{q_{\boldsymbol{w}}(\mathbf{D}|\mathbf{L}, \mathbf{\Phi}, \mathbf{A})}[\log p_{\boldsymbol{\eta}}(\mathbf{L}, \mathbf{\Phi}, \mathbf{A}|\mathbf{D})] = \frac{1}{M} \sum_{m=1}^{M} &\left( \mathrm{tr}\left((\mathbf{I} - \mathbf{L})^\top \log\left(\mathbf{I} - \mathbf{T}^{(m)}\right)\right) \right. \\ &\left. + \mathrm{tr}\left(\mathbf{L}^\top \log \mathbf{T}^{(m)}\right) - \|\mathbf{A} - S\left(\mathbf{D}^{(m)} \mathbf{D}^{(m)\top}\right)\|_F^2 \right). \end{aligned} \tag{9}$$

---

[1] More detailed calculations can be seen in Appendix A.1.

**Algorithm 1** VALEN Algorithm

---

**Input:** The PLL training set $\mathcal{D} = \{(\boldsymbol{x}_i, S_i)\}_{i=1}^n$, epoch $T$ and iteration $I$;
1: Initialize the predictive model $\boldsymbol{\theta}$ by warm-up training, the reference model $\boldsymbol{w}$ and observation model $\boldsymbol{\eta}$;
2: Extract the features $\boldsymbol{\Phi}$ from predictive model $\boldsymbol{\theta}$ and calculate the adjacency matrix $\mathbf{A}$;
3: **for** $t = 1, \ldots, T$ **do**
4:     Shuffle training set $\mathcal{D} = \{(\boldsymbol{x}_i, S_i)\}_{i=1}^n$ into $I$ mini-batches;
5:     **for** $k = 1, \ldots, I$ **do**
6:         Obtain label distribution $\boldsymbol{d}_i$ for each example $\boldsymbol{x}_i$ by Eq. (4);
7:         Update $\boldsymbol{\theta}$, $\boldsymbol{w}$ and $\boldsymbol{\eta}$ by forward computation and back-propagation by fusing Eq. (12) and Eq. (13);
8:     **end for**
9: **end for**
**Output:** The predictive model $\boldsymbol{\theta}$.

---

Note that we can use only one MC sample in Eq. (9) during the training process as suggested in [19, 35].

In addition, VALEN improves the label enhancement by employing the compatibility loss, which enforces that the recovered label distributions should not be completely different from the confidence $\zeta(\boldsymbol{x}_i)$ [9, 26] estimated by current prediction $f(\boldsymbol{x}_i; \boldsymbol{\theta})$:

$$\mathcal{L}_o = -\frac{1}{n} \sum_{i=1}^n \sum_{j=1}^c \zeta_j(\boldsymbol{x}_i) \log d_i^{y_j} \tag{10}$$

where

$$\zeta_j(\boldsymbol{x}_i) = \begin{cases} f_j(\boldsymbol{x}_i; \boldsymbol{\theta}) / \sum_{y_k \in S_i} f_k(\boldsymbol{x}_i; \boldsymbol{\theta}) & \text{if } y_j \in S_i \\ 0 & \text{otherwise} \end{cases} \tag{11}$$

Now we can easily get the objective of label enhancement $\mathcal{L}_{LE}$ as follows:

$$\mathcal{L}_{LE} = \lambda \mathcal{L}_o - \mathcal{L}_{ELBO} \tag{12}$$

where $\lambda$ is a hyper-parameter. The label distribution matrix $\mathbf{D}$ is sampled from $q(\mathbf{D}|\mathbf{L}, \boldsymbol{\Phi}, \mathbf{A})$, i.e., $\boldsymbol{d}_i \sim Dir(\boldsymbol{\alpha}_i)$. Note that the implicit reparameterization gradient [10] is employed, which avoids the inversion of the standardization function, which makes the gradients can be computed analytically in backward pass.

## 2.4 Classifier Training

To train the predictive model, we minimize the following empirical risk estimator by levering the recovered label distributions:

$$\widehat{R}_V(f) = \frac{1}{n} \sum_{i=1}^n \left( \sum_{y_j \in S_i} \frac{d_i^{y_j}}{\sum_{y_j \in S_i} d_i^{y_j}} \ell(f(\boldsymbol{x}_i), \boldsymbol{e}^{y_j}) \right). \tag{13}$$

Here we adopt the average value of $\boldsymbol{d}_i$ sampled by $\boldsymbol{d}_i \sim Dir(\boldsymbol{\alpha}_i)$. We can use any deep neural network as the predictive model, and then equip it with the VALEN framework to deal with PLL. Note that we could train the predictive model and update the label distributions in a principled end-to-end manner by fusing the objective Eq. (12) and Eq. (13). The algorithmic description of the VALEN is shown in Algorithm 1.

Let $\widehat{f}_V = \min_{f \in \mathcal{F}} \widehat{R}_V(f)$ be the empirical risk minimizer and $f^\star = \min_{f \in \mathcal{F}} R_V(f)$ be the optimal risk minimizer where $R_V(f)$ is the risk estimator. Besides, we define the function space $\mathcal{H}_{y_j}$ for the label $y_j \in \mathcal{Y}$ as $\{h : \boldsymbol{x} \mapsto f_{y_j}(\boldsymbol{x}) \mid f \in \mathcal{F}\}$. Let $\mathfrak{R}_n(\mathcal{H}_{y_j})$ be the expected Rademacher complexity [2] of $\mathcal{H}_{y_j}$ with sample size $n$, then we have the following theorem.

**Theorem 1** *Assume the loss function $\ell(f(\boldsymbol{x}), \boldsymbol{e}^{y_j})$ is L-Lipschitz with respect to $f(\boldsymbol{x})(0 < L < \infty)$ for all $y_j \in \mathcal{Y}$ and upper-bounded by $M$, i.e., $M = \sup_{x \in \mathcal{X}, f \in \mathcal{F}, y_j \in \mathcal{Y}} \ell(f(x), \boldsymbol{e}^{y_j})$. Then, for any*

$\delta > 0$, *with probability at least* $1 - \delta$,

$$R\left(\widehat{f}_V\right) - R\left(f^\star\right) \le 4\sqrt{2}L \sum_{j=1}^{c} \mathfrak{R}_n\left(\mathcal{H}_{y_j}\right) + M\sqrt{\frac{\log\frac{2}{\delta}}{2n}}$$

The proof of Theorem 1 is provided in Appendix A.2. Theorem 1 shows that the empirical risk minimizer $f_V$ converges to the optimal risk minimizer $f^\star$ as $n \to \infty$ and $\mathfrak{R}_n\left(\mathcal{H}_{y_j}\right) \to 0$ for all parametric models with a bounded norm.

## 3 Related Work

As shown in Section 1, supervision information conveyed by partially labeled training examples is implicit as the ground-truth label is hidden within the candidate label set. Therefore, partial label learning can be regarded as a *weak supervision* learning framework [18] with implicit labeling information. Intuitively, the basic strategy for handling partial label learning is disambiguation, i.e., trying to identify the ground-truth label from the candidate label set associated with each training example, where existing strategies include disambiguation by identification or disambiguation by averaging. For identification-based disambiguation, the ground-truth label is regarded as a latent variable and identified [17, 28, 24, 5, 37]. For averaging-based disambiguation, all the candidate labels are treated equally and the prediction is made by averaging their modeling outputs [15, 7, 40].

Most existing algorithms aim to fulfill the learning task by fitting widely-used learning techniques to partial label data. For maximum likelihood techniques, the likelihood of observing each partially labeled training example is defined over its candidate label set instead of the unknown ground-truth label [17, 24]. $K$-nearest neighbor techniques determine the class label of unseen instances via voting among the candidate labels of its neighboring examples [15, 40]. For maximum margin techniques, the classification margins over the partially labeled training examples are defined by discriminating modeling outputs from candidate labels and non-candidate labels [28, 37]. For boosting techniques, the weight over each partially labeled training example and the confidence over the candidate labels are updated in each boosting round [31]. For disambiguation-free strategies, the generalized description degree is estimated by using a graph Laplacian and induce a multi-output regression [34]. The confidence of each candidate label is estimated by using the manifold structure of feature space [42]. However, these methods just estimate the soft labeling information and train the predictive models in separate stages without considering the feedback of the predictive models.

The above-mentioned works were solved in specific low-efficiency manners and incompatible with high-efficient stochastic optimization. To handle large-scale datasets, the deep networks are employed with an entropy-based regularizer to maximize the margin between the potentially correct label and the unlikely ones [36]. [26] proposes a classifier-consistent risk estimator and a progressive identification, which is compatible with deep models and stochastic optimizers. [9] proposes a statistical model to depict the generation process of candidate label sets, which deduces a risk-consistent method and a classifier-consistent method.

The previous methods assume that the candidate labels are randomly sampled with the uniform generating procedure. However, the candidate labels are always instance-dependent (feature-dependent) in practice as the incorrect labels related to the feature are more likely to be picked as candidate label set for each instance. In this paper, we consider instance-dependent PLL and assume that each instance in PLL is associated with a latent *label distribution* [33, 35, 11] constituted by the real number of each label, representing the degree to each label describing the feature. Label enhancement (LE) [12, 33, 35] recovers the latent label distribution from the observed logical labels, in which the recovered label distribution is a kind of pseudo label [29, 23], actually. Note that the generation process of soft labels in label smoothing [30, 27] and distillation [14, 39] could also be regarded as a label enhancement process.

Table 1: Classification accuracy (mean±std) of each comparing approach on benchmark datasets corrupted by the instance-dependent generating procedure.

|  | MNIST | Kuzushiji-MNIST | Fashion-MNIST | CIFAR-10 |
|---|---|---|---|---|
| VALEN | **97.85±0.05%** | **86.19±0.14%** | **86.17±0.19%** | **80.38±0.52%** |
| PRODEN | 97.69±0.04%• | 85.71±0.12%• | 85.54±0.09%• | 79.80±0.28%• |
| RC | 97.60±0.05%• | 84.86±0.11%• | 85.51±0.10%• | 79.46±0.25%• |
| CC | 97.44±0.03%• | 82.67±1.82%• | 85.19±0.04%• | 78.98±0.60%• |
| D2CNN | 94.63±0.16%• | 83.03±0.78%• | 82.42±0.21%• | 73.11±0.11%• |
| GA | 95.25±0.07%• | 82.45±0.63%• | 80.41±0.24%• | 77.57±0.76%• |

Table 2: Classification accuracy (mean±std) of each comparing approach on benchmark datasets corrupted by the uniform generating procedure.

|  | MNIST | Kuzushiji-MNIST | Fashion-MNIST | CIFAR-10 |
|---|---|---|---|---|
| VALEN | 97.93±0.05% | **88.76±0.26%** | **88.98±0.16%** | **81.93±1.01%** |
| PRODEN | **97.97±0.03%** | 88.55±0.10% | 88.94±0.12% | 81.53±0.53% |
| RC | 97.86±0.03% | 86.65±0.10%• | 88.59±0.08%• | 81.30±1.30% |
| CC | 97.73±0.02%• | 87.99±0.03%• | 88.93±0.06% | 80.17±1.09%• |
| D2CNN | 95.12±0.16%• | 84.03±0.78%• | 80.42±0.21%• | 75.11±0.11%• |
| GA | 96.29±0.19%• | 82.36±0.98%• | 81.81±0.99%• | 60.14±1.35%• |

## 4 Experiments

### 4.1 Datasets

We adopt four widely used benchmark datasets including `MNIST` [22], `Fashion-MNIST` [32], `Kuzushiji-MNIST` [6], and `CIFAR-10` [21], and five datasets from the UCI Machine Learning Repository [1], including `Yeast`, `Texture`, `Dermatology`, `Synthetic Control`, and `20Newgroups`.

We manually corrupt these datasets into partially labeled versions [2] by using a flipping probability $\xi_i^{y_j} = P(l_i^{y_j} = 1|\hat{l}_i^{y_j} = 0, \boldsymbol{x}_i)$, where $\hat{l}_i^{y_j}$ is the original clean label. To synthesize the instance-dependent candidate labels, we set the flipping probability of each incorrect label corresponding to an example $\boldsymbol{x}_i$ by using the confidence prediction of a clean neural network $\hat{\boldsymbol{\theta}}$ (trained with the original clean labels) [43] with $\xi_i^{y_j} = \frac{f_j(\boldsymbol{x}_i;\hat{\boldsymbol{\theta}})}{\max_{y_j \in \bar{Y}_i} f_j(\boldsymbol{x}_i;\hat{\boldsymbol{\theta}})}$, where $\bar{Y}_i$ is the incorrect label set of $\boldsymbol{x}_i$. The uniform corrupted version adopts the uniform generating procedure [26, 9] to flip the incorrect label into candidate label, where $\xi_i^{y_j} = \frac{1}{2}$.

In addition, five real-world PLL datasets are adopted, which are collected from several application domains including `Lost` [7], `Soccer Player` [38] and `Yahoo!News` [13] for automatic face naming from images or videos, `MSRCv2` [24] for object classification, and `BirdSong` [3] for bird song classification. The detailed descriptions of these datasets are provided in Appendix A.3.

We run 5 trials on the four benchmark datasets and perform five-fold cross-validation on UCI datasets and real-world PLL datasets. The mean accuracy as well as standard deviation are recorded for all comparing approaches.

### 4.2 Baselines

The performance of VALEN is compared against five DNN based approaches: 1) PRODEN [26]: A progressive identification partial label learning approach which approximately minimizes a risk estimator and identifies the true labels in a seamless manner; 2) RC [9]: A risk-consistent partial label learning approach which employs the importance reweighting strategy to converges the true risk

---

[2]The datasets corrupted by the instance-dependent generating procedure are available at `https://drive.google.com/drive/folders/1J_68EqOrLN6tA56RcyTgcr1komJB31Y1?usp=sharing`.

Table 3: Classification accuracy (mean±std) of each comparing approach on UCI datasets corrupted by the instance-dependent generating procedure.

| | Yeast | Texture | Synthetic Control | Dermatology | 20Newsgroup |
|---|---|---|---|---|---|
| VALEN | **57.57±1.08%** | **94.76±0.93%** | **82.86±0.76%** | 89.86±1.31% | **81.88±0.47%** |
| PRODEN | 54.78±1.28%● | 89.87±2.14%● | 71.16±6.19%● | 88.53±3.87% | 78.06±0.74%● |
| RC | 54.77±1.27%● | 89.57±2.37%● | 65.99±2.72%● | 88.53±3.49% | 78.02±0.79%● |
| CC | 54.98±0.91%● | 88.92±7.56% | 66.99±2.47%● | 88.26±4.11% | 77.88±0.39%● |
| D2CNN | 44.94±1.87%● | 69.52±5.79%● | 62.66±8.92%● | 81.95±6.18%● | 73.55±0.92%● |
| GA | 25.86±3.17%● | 74.84±2.87%● | 56.43±1.29%● | 84.85±1.43%● | 49.49±3.42%● |
| CLPL | 54.92±2.38% | 81.27±9.09%● | 66.33±3.25%● | 92.07±3.42% | 77.62±0.23%● |
| PL-SVM | 41.85±5.92%● | 39.03±4.35%● | 50.33±5.73%● | 84.98±4.56% | 72.89±0.41%● |
| PL-KNN | 47.44±2.69%● | 70.05±0.70%● | 80.50±1.26%● | 83.61±3.15%● | 33.28±1.09%● |
| IPAL | 56.40±2.07% | 93.49±0.89% | 77.66±3.60%● | 78.94±8.34%● | 67.38±0.95%● |
| PLLE | 55.53±1.74% | 84.45±1.07%● | 66.16±7.96%● | **93.16±2.58%**○ | 75.54±0.66%● |

Table 4: Classification accuracy (mean±std) of each comparing approach on UCI datasets corrupted by the uniform generating procedure.

| | Yeast | Texture | Synthetic Control | Dermatology | 20Newsgroup |
|---|---|---|---|---|---|
| VALEN | **58.18±1.46%** | 97.30±0.57% | **97.17±0.47%** | **97.07±0.41%** | **71.75±3.02%** |
| PRODEN | 56.32±1.98% | 97.75±0.53% | 95.83±1.95% | 95.07±1.84%● | 68.28±0.91%● |
| RC | 56.39±1.85% | 97.77±0.55% | 95.99±1.80% | 95.62±1.51% | 68.44±1.09% |
| CC | 56.25±1.89% | 97.79±0.57% | 96.33±1.39% | 95.90±1.69% | 67.95±0.95%● |
| D2CNN | 54.04±1.90%● | 97.23±0.72% | 81.16±8.11%● | 90.43±2.38%● | 65.88±2.56%● |
| GA | 22.98±2.57%● | 95.09±1.07%● | 56.87±1.53%● | 51.95±3.89%● | 58.29±1.74%● |
| CLPL | 56.54±3.35% | 98.14±0.59% | 94.66±6.41% | 96.72±0.76% | 70.45±0.91% |
| PL-SVM | 46.23±7.21%● | 39.74±2.11%● | 76.50±5.31%● | 92.37±5.08% | 70.44±0.37% |
| PL-KNN | 44.40±2.50%● | 95.31±0.85%● | 95.33±2.98% | 92.91±2.92%● | 27.10±0.49%● |
| IPAL | 43.86±3.39%● | **98.71±0.37%**○ | 96.83±1.90% | 95.35±2.08% | 65.39±1.21%● |
| PLLE | 53.58±2.86%● | 98.40±0.40%○ | 89.66±1.91%● | 90.98±1.85%● | 53.88±0.59%● |

minimizer; 3) CC [9]: A classifier-consistent partial label learning approach which uses a transition matrix to form an empirical risk estimator; 4) D2CNN [36]: A deep partial label learning approach which design an entropy-based regularizer to maximize the margin between the potentially correct label and the unlikely ones; 5) GA [16]: An unbiased risk estimator approach which can be applied for partial label learning.

For all the DNN based approaches, we adopt the same predictive model for fair comparisons. Specifically, the 32-layer ResNet is trained on CIFAR-10 in which the learning rate, weight decay and mini-batch size are set to $0.05$, $10^{-3}$ and 256, respectively. The three-layer MLP is trained on MNIST, Fashion-MNIST and Kuzushiji-MNIST where the learning rate, weight decay and mini-batch size are set to $10^{-2}$, $10^{-4}$ and 256, respectively. The linear model is trained on UCI and real-world PLL datasets where the learning rate, weight decay and mini-batch size are set to $10^{-2}$, $10^{-4}$ and 100, respectively. We implement the comparing methods with PyTorch. The number of epochs is set to 500, in which the first 10 epochs are warm-up training.

In addition, we also compare with five classical partial label learning approaches, each configured with parameters suggested in respective literatures: 1) CLPL [7]: A convex partial label learning approach which uses averaging-based disambiguation; 2) PL-KNN [15]: An instance-based partial label learning approach which works by $k$-nearest neighbor weighted voting; 3) PL-SVM [28]: A maximum margin partial label learning approach which works by identification-based disambiguation; 4) IPAL [41]: A non-parametric method that applies the label propagation strategy to iteratively update the confidence of each candidate label; 5) PLLE [34]: A two-stage partial label learning approach which estimates the generalized description degree of each class label values via graph Laplacian and induces a multi-label predictive model with the generalized description degree in separate stages.

Table 5: Classification accuracy (mean±std) of each comparing approach on the real-world datasets.

|  | Lost | MSRCv2 | BirdSong | Soccer Player | Yahoo!News |
|---|---|---|---|---|---|
| VALEN | 70.28±2.29% | 47.61±1.79% | **72.02±0.37%** | **55.90±0.58%** | **67.52±0.19%** |
| PRODEN | 68.62±4.86% | 44.47±2.33% | 71.68±0.83% | 54.40±0.85%● | 67.12±0.97% |
| RC | 68.89±5.02% | 44.59±2.65% | 71.56±0.88% | 54.23±0.89%● | 67.04±0.88% |
| CC | 62.21±1.77%● | 47.49±2.31% | 68.42±0.99%● | 53.50±0.96%● | 61.92±0.96%● |
| D2CNN | 68.56±6.68% | 43.27±2.98%● | 65.48±2.57%● | 48.16±0.62%● | 52.46±1.71%● |
| GA | 50.21±3.62%● | 30.91±4.31%● | 34.57±3.41%● | 50.65±0.94%● | 45.72±1.75%● |
| CLPL | **74.15±3.03%** | 44.47±2.58% | 65.76±1.19%● | 50.01±1.03%● | 53.25±1.12%● |
| PL-SVM | 71.56±2.71% | 38.25±3.89%● | 50.66±4.23%● | 36.39±1.03%● | 51.24±0.72%● |
| PL-KNN | 33.87±2.48%● | 43.28±2.35%● | 64.34±0.75%● | 49.24±1.23%● | 40.38±0.37%● |
| IPAL | 72.10±2.75% | **52.96±1.36%**○ | 70.32±0.91%● | 54.41±0.68%● | 66.04±0.85%● |
| PLLE | 72.55±3.55% | 47.54±1.96% | 70.63±1.24%● | 53.38±1.03%● | 59.45±0.43%● |

### 4.3 Experimental Results

Table 1 reports the classification accuracy of each DNN-based method on benchmark datasets corrupted by the instance-dependent generating procedure. The best results are highlighted in bold. In addition, ● / ○ indicates whether VALEN is statistically superior/inferior to the comparing approach on each dataset (pairwise $t$-test at 0.05 significance level). From the table, we can observe that VALEN always achieves the best performance and significantly outperforms other compared methods in most cases. In addition, we also validate the effectiveness of our approach on uniform corrupted versions that is commonly adopted in previous works. From Table 2, we can observe that VALEN achieves superior or at least comparable performance to other approaches on uniform corrupted versions.

Table 3 and Table 4 report the classification accuracy of each method on UCI datasets corrupted by the instance-dependent generating procedure and the uniform generating procedure, respectively. VALEN always achieves the best performance and significantly outperforms other DNN-based methods in most cases on instance-dependent corrupted versions while achieves superior or at least comparable performance to other approaches on uniform corrupted versions. We further compare VALEN with five classical PLL methods that can hardly be implemented by DNNs on large-scale datasets. Despite the small scale of most UCI datasets, VALEN always achieve the best performance in most cases against the classical PLL methods as VALEN can deal with the high average number of candidate labels (can be seen in Appendix A.3) in the corrupted UCI datasets.

Table 5 reports the experimental results on real-world PLL datasets. We can find that VALEN achieves best performance against other DNN-based methods on the real-world PLL datasets. Note that VALEN achieves best performance against classical methods on all datasets except `Lost` and `MSRCv2` as these datasets are small-scale and the average number of candidate labels in each dataset is low (can be seen in Appendix A.3), which leads to the result that DNN-based methods cannot take full advantage.

Figure 2(a) and Figure 2(b) illustrate the performance of VALEN on `KMNIST` corrupted by the instance-dependent generating procedure and the uniform generating procedure under different flipping probability, respectively. Besides, the performance of the ablation version that removes the label enhancement and trains the predictive model with PLL minimal loss (denoted by VALEN-NON) is recorded. These results clearly validate the usefulness of recovered label distributions for improving predictive performance. Figure 2(c) illustrates the recovered label distribution matrix over all training examples converges as the number of epoch (after warm-up training) on `Kuzushiji-MNIST`. We can see that the recovered label distributions converge fast with the increasing number of epoch.

## 5 Conclusion

In this paper, the problem of partial label learning is studied where a novel approach VALEN is proposed. We for the first time consider the instance-dependent PLL and assume that each partially labeled example is associated with a latent label distribution, which is the essential labeling information and worth being recovered for predictive model training. VALEN recovers the latent label distribution via inferring the true posterior density of the latent label distribution by Dirichlet density

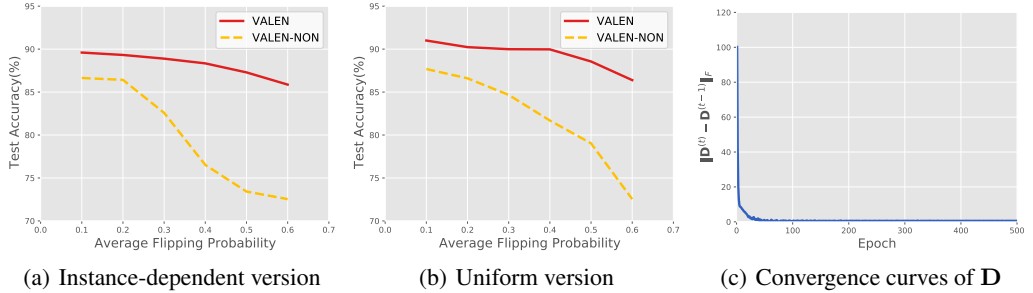

| (a) Instance-dependent version | (b) Uniform version | (c) Convergence curves of $\mathbf{D}$ |

Figure 2: Further analysis of VALEN on KMNIST.

parameterized with an inference model and deduce the evidence lower bound for optimization. In addition, VALEN iteratively recovers latent label distributions and trains the predictive model in every epoch. The effectiveness of the proposed approach is validated via comprehensive experiments on both synthesis datasets and real-world PLL datasets.

## 6 Acknowledgments

This research was supported by the National Key Research & Development Plan of China (No. 2018AAA0100104, No. 2018AAA0100100), the National Science Foundation of China (62125602, 62076063), China Postdoctoral Science Foundation (2021M700023), Jiangsu Province Science Foundation for Youths (BK20210220).

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
