# Instance-Dependent Partial Label Learning

**Ning Xu, Congyu Qiao, Xin Geng,**[*] **and Min-Ling Zhang**
School of Computer Science and Engineering, Southeast University, Nanjing 210096, China
MOE Key Laboratory of Computer Network and Information Integration, China
{xning, qiaocy, xgeng, zhangml}@seu.edu.cn

## A   Appendix

### A.1   Calculation Details of Eq. (5)

$$\log p(\mathbf{L}, \mathbf{\Phi}, \mathbf{A}) = \log p(\mathbf{D}, \mathbf{L}, \mathbf{\Phi}, \mathbf{A}) - \log p(\mathbf{D} \mid \mathbf{L}, \mathbf{\Phi}, \mathbf{A}) \tag{1}$$

Multiply both sides by $q_{\boldsymbol{w}}(\mathbf{D} \mid \mathbf{L}, \mathbf{\Phi}, \mathbf{A})$, and for $\mathbf{D}$ integral:

$$\int_{\mathbf{D}} q_{\boldsymbol{w}}(\mathbf{D} \mid \mathbf{L}, \mathbf{\Phi}, \mathbf{A}) \log p(\mathbf{L}, \mathbf{\Phi}, \mathbf{A}) d\mathbf{D} = \int_{\mathbf{D}} q_{\boldsymbol{w}}(\mathbf{D} \mid \mathbf{L}, \mathbf{\Phi}, \mathbf{A})(\log p(\mathbf{D}, \mathbf{L}, \mathbf{\Phi}, \mathbf{A}) \\ - \log p(\mathbf{D} \mid \mathbf{L}, \mathbf{\Phi}, \mathbf{A})) d\mathbf{D}. \tag{2}$$

On the left side, $\log p(\mathbf{L}, \mathbf{\Phi}, \mathbf{A})$ is independent of $\mathbf{D}$:

$$\begin{aligned}
\log p(\mathbf{L}, \mathbf{\Phi}, \mathbf{A}) &= \int_{\mathbf{D}} q_{\boldsymbol{w}}(\mathbf{D} \mid \mathbf{L}, \mathbf{\Phi}, \mathbf{A})(\log p(\mathbf{D}, \mathbf{L}, \mathbf{\Phi}, \mathbf{A}) - \log p(\mathbf{D} \mid \mathbf{L}, \mathbf{\Phi}, \mathbf{A})) d\mathbf{D} \\
&= \int_{\mathbf{D}} q_{\boldsymbol{w}}(\mathbf{D} \mid \mathbf{L}, \mathbf{\Phi}, \mathbf{A})(\log \frac{p(\mathbf{D}, \mathbf{L}, \mathbf{\Phi}, \mathbf{A})}{q_{\boldsymbol{w}}(\mathbf{D} \mid \mathbf{L}, \mathbf{\Phi}, \mathbf{A})} - \log \frac{p(\mathbf{D} \mid \mathbf{L}, \mathbf{\Phi}, \mathbf{A})}{q_{\boldsymbol{w}}(\mathbf{D} \mid \mathbf{L}, \mathbf{\Phi}, \mathbf{A})}) d\mathbf{D} \\
&= \int_{\mathbf{D}} q_{\boldsymbol{w}}(\mathbf{D} \mid \mathbf{L}, \mathbf{\Phi}, \mathbf{A})(\log \frac{p(\mathbf{D}, \mathbf{L}, \mathbf{\Phi}, \mathbf{A})}{q_{\boldsymbol{w}}(\mathbf{D} \mid \mathbf{L}, \mathbf{\Phi}, \mathbf{A})} d\mathbf{D} \\
&\quad + \text{KL}\left[q_w(\mathbf{D} \mid \mathbf{L}, \mathbf{\Phi}, \mathbf{A}) \| p(\mathbf{D} \mid \mathbf{L}, \mathbf{\Phi}, \mathbf{A})\right].
\end{aligned} \tag{3}$$

On the right side, the first term is called ELBO:

$$\begin{aligned}
\mathcal{L}_{ELBO} &= \int_{\mathbf{D}} q_{\boldsymbol{w}}(\mathbf{D} \mid \mathbf{L}, \mathbf{\Phi}, \mathbf{A})(\log \frac{p(\mathbf{D}, \mathbf{L}, \mathbf{\Phi}, \mathbf{A})}{q_{\boldsymbol{w}}(\mathbf{D} \mid \mathbf{L}, \mathbf{\Phi}, \mathbf{A})} d\mathbf{D} \\
&= \int_{\mathbf{D}} q_{\boldsymbol{w}}(\mathbf{D} \mid \mathbf{L}, \mathbf{\Phi}, \mathbf{A})(\log \frac{p(\mathbf{D}) p(\mathbf{L}, \mathbf{\Phi}, \mathbf{A} \mid \mathbf{D})}{q_{\boldsymbol{w}}(\mathbf{D} \mid \mathbf{L}, \mathbf{\Phi}, \mathbf{A})} d\mathbf{D} \\
&= \mathbb{E}_{q_w(\mathbf{D}|\mathbf{L}, \mathbf{\Phi}, \mathbf{A})}\left[\log p_{\boldsymbol{\eta}}(\mathbf{L}, \mathbf{\Phi}, \mathbf{A} \mid \mathbf{D})\right] - \text{KL}\left[q_{\boldsymbol{w}}(\mathbf{D} \mid \mathbf{L}, \mathbf{\Phi}, \mathbf{A}) \| p(\mathbf{D})\right].
\end{aligned} \tag{4}$$

Then we have

$$\log p(\mathbf{L}, \mathbf{\Phi}, \mathbf{A}) = \mathcal{L}_{ELBO} + \text{KL}\left[q_w(\mathbf{D} \mid \mathbf{L}, \mathbf{\Phi}, \mathbf{A}) \| p(\mathbf{D} \mid \mathbf{L}, \mathbf{\Phi}, \mathbf{A})\right]. \tag{5}$$

### A.2   Proofs of Theorem 1

**Definition 1** *Let $Z_1, \ldots, Z_n$ be $n$ i.i.d. random variables drawn from a probability distribution $\mu, \mathcal{H} = \{h : \mathcal{Z} \to \mathbb{R}\}$ be a class of measurable functions. Then the expected Rademacher complexity*

---

[*]Corresponding author

35th Conference on Neural Information Processing Systems (NeurIPS 2021).

*of $\mathcal{H}$ is defined as*

$$\mathfrak{R}_n(\mathcal{H}) = \mathbb{E}_{Z_1,\ldots,Z_n \sim \mu}\mathbb{E}_\sigma\left[\sup_{h\in\mathcal{H}}\frac{1}{n}\sum_{i=1}^{n}\sigma_i h\left(Z_i\right)\right]$$

*where $\boldsymbol{\sigma} = (\sigma_1,\ldots,\sigma_n)$ are Rademacher variables taking the value from $\{-1,+1\}$ with even probabilities.*

The risk estimator in Eq. (12) can be rewritten as:

$$\widehat{R}_V(f) = \frac{1}{n}\sum_{i=1}^{n}\sum_{j=1}^{c}\chi_i^j\ell(f(\boldsymbol{x}_i), \boldsymbol{e}^{y_j}). \tag{6}$$

where $\chi_i^j = \frac{d_i^{y_j}}{\sum_{y_j\in S_i}d_i^{y_j}}$ if $y_j \in S_i$ and $\chi_i^j = 0$ otherwise. Then we define a function space as:

$$\mathcal{G}_{\mathrm{V}} = \left\{(\boldsymbol{x}, S) \mapsto \sum_{j=1}^{c}\chi^j\ell(f(\boldsymbol{x}), \boldsymbol{e}^{y_j}) \mid f \in \mathcal{F}\right\} \tag{7}$$

Let $\widetilde{\mathfrak{R}}_n\left(\mathcal{G}_V\right)$ be the expected Rademacher complexity of $\mathcal{G}_V$, i.e.

$$\widetilde{\mathfrak{R}}_n\left(\mathcal{G}_V\right) = \mathbb{E}_{p(\boldsymbol{x},S)}\mathbb{E}_{\boldsymbol{\sigma}}\left[\sup_{g\in\mathcal{G}_V}\frac{1}{n}\sum_{i=1}^{n}\sigma_i g\left(\boldsymbol{x}_i, S_i\right)\right]. \tag{8}$$

Then we have

**Lemma 1** *Suppose the loss function $\ell$ is bounded by $M$, i.e., $M = \sup_{\boldsymbol{x}\in\mathcal{X}, f\in\mathcal{F}, y_j\in\mathcal{Y}}\ell(f(\boldsymbol{x}), y)$, then for any $\delta > 0$, with probability at least $1 - \delta$,*

$$\sup_{f\in\mathcal{F}}\left|R_V(f) - \widehat{R}_V(f)\right| \leq 2\widetilde{\mathfrak{R}}_n\left(\mathcal{G}_V\right) + \frac{M}{2}\sqrt{\frac{\log\frac{2}{\delta}}{2n}}.$$

*Proof.* In order to prove this lemma, we first show that the one direction $\sup_{f\in\mathcal{F}}R_V(f) - \widehat{R}_V(f)$ is bounded with probability at least $1 - \delta/2$, and the other direction can be similarly shown. Suppose an example $(\boldsymbol{x}_i, S_i)$ is replaced by another arbitrary example $(\boldsymbol{x}_i', S_i')$, then the change of $\sup_{f\in\mathcal{F}}R_V(f) - \widehat{R}_V(f)$ is no greater than $M/(2n)$, since $\ell$ is bounded by $M$. By applying McDiarmid's inequality [9], for any $\delta > 0$, with probability at least $1 - \delta/2$,

$$\sup_{f\in\mathcal{F}}R_V(f) - \widehat{R}_V(f) \leq \mathbb{E}\left[\sup_{f\in\mathcal{F}}R_V(f) - \widehat{R}_V(f)\right] + \frac{M}{2}\sqrt{\frac{\log\frac{2}{\delta}}{2n}} \tag{9}$$

By symmetrization [10], we can obtain

$$\mathbb{E}\left[\sup_{f\in\mathcal{F}}R_V(f) - \widehat{R}_V(f)\right] \leq 2\widetilde{\mathfrak{R}}_n\left(\mathcal{G}_V\right) \tag{10}$$

By further taking into account the other side $\sup_{f\in\mathcal{F}}\widehat{R}_V(f) - R_V(f)$, we have for any $\delta > 0$, with probability at least $1 - \delta$,

$$\sup_{f\in\mathcal{F}}\left|R_V(f) - \widehat{R}_V(f)\right| \leq 2\widetilde{\mathfrak{R}}_n\left(\mathcal{G}_V\right) + \frac{M}{2}\sqrt{\frac{\log\frac{2}{\delta}}{2n}}. \tag{11}$$

**Lemma 2** *Assume the loss function $\ell(f(\boldsymbol{x}), \boldsymbol{e}^{y_j})$ is L-Lipschitz with respect to $f(\boldsymbol{x})(0 < L < \infty)$ for all $y_j \in \mathcal{Y}$. Then, the following inequality holds:*

$$\widetilde{\mathfrak{R}}_n\left(\mathcal{G}_V\right) \leq \sqrt{2}L\sum_{j=1}^{c}\mathfrak{R}_n\left(\mathcal{H}_{y_j}\right)$$

Table 1: Characteristic of the benchmark datasets.

| Dataset | #Train | #Test | #Features | #Class Labels | avg. #CLs_U | avg. #CLs_F |
|---|---|---|---|---|---|---|
| **MNIST** | 60,000 | 10,000 | 784 | 10 | 5.50 | 4.94 |
| **Fashion-MNIST** | 60,000 | 10,000 | 784 | 10 | 5.51 | 4.61 |
| **Kuzushiji-MNIST** | 60,000 | 10,000 | 784 | 10 | 5.49 | 4.34 |
| **CIFAR-10** | 50,000 | 10,000 | 3,072 | 10 | 5.49 | 2.74 |
| **Yeast** | 1,187 | 297 | 8 | 10 | 5.54 | 2.83 |
| **Texture** | 4,400 | 1,100 | 40 | 11 | 5.99 | 2.52 |
| **Synthetic Control** | 480 | 120 | 60 | 6 | 3.59 | 2.27 |
| **Dermatology** | 293 | 73 | 34 | 6 | 3.54 | 2.35 |
| **20Newsgroups** | 15,076 | 3,770 | 300 | 20 | 10.48 | 3.36 |

Table 2: Characteristic of the real-world PLL datasets.

| Dataset | #Train | #Test | #Features | #Class Labels | avg. #CLs | Task Domain |
|---|---|---|---|---|---|---|
| **Lost** | 898 | 224 | 108 | 16 | 2.23 | *automatic face naming* [4] |
| **MSRCv2** | 1,406 | 352 | 48 | 23 | 3.16 | *object classification* [8] |
| **BirdSong** | 3,998 | 1000 | 38 | 13 | 2.18 | *bird song classification* [2] |
| **Soccer Player** | 13,978 | 3,494 | 279 | 171 | 2.09 | *automatic face naming* [12] |
| **Yahoo! News** | 18,393 | 4,598 | 163 | 219 | 1.91 | *automatic face naming* [5] |

*where*

$$\mathcal{H}_{y_j} = \left\{ h : \boldsymbol{x} \mapsto f_{y_j}(\boldsymbol{x}) \mid f \in \mathcal{F} \right\},$$

$$\mathfrak{R}_n \left( \mathcal{H}_{y_j} \right) = \mathbb{E}_{p(\boldsymbol{x})} \mathbb{E}_{\boldsymbol{\sigma}} \left[ \sup_{h \in \mathcal{H}_{y_j}} \frac{1}{n} \sum_{i=1}^{n} h\left(\boldsymbol{x}_i\right) \right]. \tag{12}$$

*Proof.* As $\chi_i^j = \frac{d_i^{y_j}}{\sum_{y_j \in S_i} d_i^{y_j}}$ if $y_j \in S_i$ and $\chi_i^j = 0$ otherwise for each example $(\boldsymbol{x}_i, S_i)$, we have $\sum_{i=j}^{c} \chi_i^j = 1$ and $\chi_i^j \in [0, 1]$. In this way, we can obtain $\widetilde{\mathfrak{R}}_n \left( \mathcal{G}_V \right) \leq \mathfrak{R}_n(\ell \circ \mathcal{F})$ where $\ell \circ \mathcal{F}$ denotes $\{\ell \circ f \mid f \in \mathcal{F}\}$. Since $\mathcal{H}_{y_j} = \left\{ h : \boldsymbol{x} \mapsto f_{y_j}(\boldsymbol{x}) \mid f \in \mathcal{F} \right\}$ and the loss function $\ell(f(\boldsymbol{x}), \boldsymbol{e}^{y_j})$ is $L$-Lipschitz with respect to $f(\boldsymbol{x})(0 < L < \infty)$ for all $y_j \in \mathcal{Y}$, by the Rademacher vector contraction inequality, we have $\mathfrak{R}_n(\ell \circ \mathcal{F}) \leq \sqrt{2}L \sum_{j=1}^{c} \mathfrak{R}_n \left( \mathcal{H}_{y_j} \right)$.

Based on Lemma 1 and 2, Theorem 1 is proven through

$$\begin{aligned}
R(\widehat{f}_V) - R\left(f^\star\right) &= R(\widehat{f}_V) - \widehat{R}_V(\widehat{f}) + \widehat{R}_V(\widehat{f}) - \widehat{R}_V\left(f^\star\right) + \widehat{R}_V\left(f^\star\right) - R\left(f^\star\right) \\
&\leq R(\widehat{f}_V) - \widehat{R}_V(\widehat{f}) + \widehat{R}_V\left(f^\star\right) - R\left(f^\star\right) \\
&\leq 2 \sup_{f \in \mathcal{F}} \left| R_V(f) - \widehat{R}_V(f) \right| \\
&\leq 4\widetilde{\mathfrak{R}}_n \left( \mathcal{G}_V \right) + M\sqrt{\frac{\log \frac{2}{\delta}}{2n}} \\
&\leq 4\sqrt{2}L \sum_{j=1}^{c} \mathfrak{R}_n \left( \mathcal{H}_{y_j} \right) + M\sqrt{\frac{\log \frac{2}{\delta}}{2n}}.
\end{aligned} \tag{13}$$

## A.3 Details of Experiments

We collect four widely used benchmark datasets including `MNIST` [7], `Fashion-MNIST` [11], `Kuzushiji-MNIST` [3], and `CIFAR-10` [6], and five datasets from the UCI Machine Learning Repository [1], including `Yeast`, `Texture`, `Dermatology`, `Synthetic Control`, and `20Newgroups`. The average number of candidate labels (avg. #CLs_F) for each corrupted dataset by instance-dependent generating procedure and he average number of candidate labels (avg. #CLs_U) for each corrupted dataset by uniform generating procedure are also recorded in Table 1.

In addition, five real-world PLL datasets are adopted, which are collected from several application domains including `Lost` [4], `Soccer Player` [12] and `Yahoo!News` [5] for automatic face naming from images or videos, `MSRCv2` [8] for object classification, and `BirdSong` [2] for bird song classification. The average number of candidate labels (avg. #CLs) for each real-world partial label data set is also recorded in Table 2.

On all the above datasets, we take the average accuracy of the last ten epochs as the accuracy for each trial. All the experiments are conducted on NVIDIA GeForce RTX 2080 GPUs.