# OpenReview forum: "Instance-Dependent Partial Label Learning"
_NeurIPS.cc/2021/Conference — NeurIPS 2021 Spotlight_

### Official Review · Reviewer_2yTr · 2021-06-28

**Rating:** 8
**Confidence:** 5

**Summary:**

This paper studies the problem of feature-dependent partial label learning, which is an interesting weakly supervised learning problem where the candidate labels of each instance are feature-dependent. Accordingly, the first attempt to feature-dependent partial label learning with latent label distribution is proposed.  The performance advantage of the proposed approach over state-of-the-art approaches is clearly validated via extensive experimental studies.

**Limitations And Societal Impact:**

1. The paper could give more discussions on the different results on the datasets corrupted by the feature-dependent generating procedure and the uniform generating procedure.
2. It seems that classical partial label learning approaches achieve better performance than DNN-based approaches on Lost and MSRCv2. The authors should give more discussion about the reasons why DNN-based methods achieve poor performance on these two datasets.


**Main Review:**

1. The problem studies in this paper, i.e., feature-dependent partial label learning, is interesting and practical since the incorrect labels related to the feature are more likely to be picked as candidate label in partially labeled data. As far as I know, this is the first study that focuses on the feature-dependent setting in partial label learning.
2. The latent label distribution developed in this paper is well motivated and clearly presented. Based on the latent label distribution, the paper recovers the latent label distributions and train the predictive model alternately.
3. This paper infers the posterior density of the latent label distribution by leveraging the approximate Dirichlet density with the evidence lower bound.
4. Comprehensive experiments are performed on synthetic as well as real-world data sets to show the effectiveness of the proposed approach.
Generally, the paper is very well organized and is easy to follow. It studied an important problem for partial label learning and proposed a novel method.


**Time Spent Reviewing:**

3

---

> ### Author Response · Authors · 2021-08-05
> **Response to Reviewer 2yTr**
>
> We are very grateful to the reviewer for this accurate summary, and for the kind recognition of our key contributions. Below we address the concerns mentioned in the review:
> 1. The paper could give more discussions on the different results on the datasets corrupted by the feature-dependent generating procedure and the uniform generating procedure.
>
> The synthetic generating procedure is designed to validate the proposed method, where the advantages are more clear with the feature-dependent generating procedure. We will add more discussion about the results on each generating procedure in the revision.
>
> 2. It seems that classical partial label learning approaches achieve better performance than DNN-based approaches on Lost and MSRCv2. The authors should give more discussion about the reasons why DNN-based methods achieve poor performance on these two datasets.
>
> Lost dataset and MSRCv2 dataset are small-scale datasets with a low ratio of candidate labels to class labels. Classical partial label learning approaches provide an advantage to deal with these datasets. We will add more analysis about the experimental results on Lost and MSRCv2 in the revision.

---

### Official Review · Reviewer_BDBH · 2021-07-14

**Rating:** 7
**Confidence:** 5

**Summary:**

This work approaches the feature-dependent partial-label learning problem where each training example has a set of candidate labels that are generated by feature-dependent procedure rather than random procedure. This paper assumes that each instance in PLL is associated with a latent label distribution which is the essential labeling information in partially labeled examples and worth being leveraged for predictive model training. Inspired by this, the authors propose a novel method to enhance the labeling information by recovering the latent label distribution and train the predictive model with recovered label distributions. In the experiment, the authors show the effectiveness of the method in the benchmark and real-world datasets.

**Limitations And Societal Impact:**

See my main review.

**Main Review:**

This paper proposes a novel partial label learning method for the feature-dependent generating procedure with latent label distribution. The example in Figure 1 motivates the problem well, which shows that the incorrect label with a high degree in the latent label distribution is more likely to be annotated as the candidate label. This is the first attempt to consider the cause of the candidate labels in feature-dependent partial label learning datasets to motivate the novel solution of solving the problem with the latent label distribution. Besides, the authors propose a posterior inference model to approximate the posterior density of latent label distribution with Dirichlet density and optimize the model with the derived evidence lower bound. The predictive network is trained on the proposed empirical risk estimator by leveraging the candidate labels as well as the label distributions.

The paper is clearly written and well organized. Comprehensive experiments are performed on synthetic as well as real-world data sets to show the effectiveness of the proposed approach. Besides, the authors conduct further experiments to validate the helpfulness of the latent label distribution for partial label learning.

Nevertheless, the literature review should be more focused on describing the works that are more related and should give more details about label distribution learning. As label distributions might be more interesting to those people working on other more common machine learning problems, the authors could give more discussion about the latent label distribution in the conclusion.

**Time Spent Reviewing:**

1.5

---

> ### Author Response · Authors · 2021-08-05
> **Response to Reviewer BDBH**
>
> We thank the reviewer for reading the paper and the positive comments about it. We will add more discussion about label distribution in the revision.

---

### Official Review · Reviewer_tK2X · 2021-07-15

**Rating:** 7
**Confidence:** 4

**Summary:**

This paper proposes an algorithm for learning from feature-dependent partial labels. The paper is based on a latent distribution over the weak labels which is estimated during training. A variational approach is used to make approximate inference computationally tractable. The experiments show that the proposed algorithms outperform many algorithm in the state of the art. Experiments are done using real data with synthetic label corruptions and also with real weak labels.

**Ethical Concerns:**

No relevant issues.

**Limitations And Societal Impact:**

A brief mention to a potential negative societal impact is included in the paper, related to the potential employment destruction as a consequence of reducing the need of human labelers.

**Main Review:**

Clarity: despite the mathematical complexity of the inference process, the main ideas of the paper are clear.

Originality: the majority of papers about PLL and, in general, weak label learning, assume (explicitly or not) that the corruption process is independent on the features. This paper is a serious attempt to overcome this limitation.

Quality: the paper is technically correct.

Significance: the experimental results show that removing the feature independent assumption, the performance of the classifiers based on weak labels can be improved. The advantages of the proposed method are more clear with the synthetic corruptions, as expected. The experiments using real-world PLL datasets are diverse, but the proposed method shows some advantage over many state of the art method in three out of five datasets.



**Time Spent Reviewing:**

4

---

> ### Author Response · Authors · 2021-08-05
> **Response to Reviewer tK2X**
>
> We thank the reviewer for the positive feedback and for highlighting the significance of our work.

---

### Decision · Program_Chairs · 2021-09-27

**Decision:**

Accept (Spotlight)

**Comment:**

This paper is the first to address partial label learning in the setting where the partial label depends on the feature vector. The authors introduce a principled approach to solve this problem and demonstrate some state-of-the-art empirical performance. This paper will be well-cited in the PLL literature and serve as an important benchmark for future research. The reviewers make a handful of comments that should be reflected in the final version. A pertinent missing reference is Katz-Samuels et al., Decontamination of Mutual Contamination Models, JMLR 2019.